# Staying out or Going in? The Interplay between Type 3 and Type 5 Secretion Systems in Adhesion and Invasion of Enterobacterial Pathogens

**DOI:** 10.3390/ijms21114102

**Published:** 2020-06-08

**Authors:** Rachel Whelan, Gareth McVicker, Jack C. Leo

**Affiliations:** Antimicrobial Resistance, Omics and Microbiota Group, Department of Biosciences, Nottingham Trent University, Nottingham NG1 4FQ, UK; rachel.whelan2015@my.ntu.ac.uk (R.W.); gareth.mcvicker@ntu.ac.uk (G.M.)

**Keywords:** autotransporter, effector protein, EHEC, type 3 secretion system, *Shigella*, *Yersinia*

## Abstract

Enteric pathogens rely on a variety of toxins, adhesins and other virulence factors to cause infections. Some of the best studied pathogens belong to the Enterobacterales order; these include enteropathogenic and enterohemorrhagic *Escherichia coli*, *Shigella* spp., and the enteropathogenic *Yersiniae*. The pathogenesis of these organisms involves two different secretion systems, a type 3 secretion system (T3SS) and type 5 secretion systems (T5SSs). The T3SS forms a syringe-like structure spanning both bacterial membranes and the host cell plasma membrane that translocates toxic effector proteins into the cytoplasm of the host cell. T5SSs are also known as autotransporters, and they export part of their own polypeptide to the bacterial cell surface where it exerts its function, such as adhesion to host cell receptors. During infection with these enteropathogens, the T3SS and T5SS act in concert to bring about rearrangements of the host cell cytoskeleton, either to invade the cell, confer intracellular motility, evade phagocytosis or produce novel structures to shelter the bacteria. Thus, in these bacteria, not only the T3SS effectors but also T5SS proteins could be considered “cytoskeletoxins” that bring about profound alterations in host cell cytoskeletal dynamics and lead to pathogenic outcomes.

## 1. Introduction

The Gram-negative families Enterobacteriaceae and Yersiniaceae are two well-studied groups of microorganisms that form part of the order Enterobacterales [1]. Both families include pathogens that are able to cause intestinal and extraintestinal infection in the human host. Intestinal infection often leads to moderate to severe diarrhoea through alteration and/or disruption of the intestinal epithelium, which in some cases results in death, and is particularly pronounced amongst children under five years old and individuals in low-income regions.

Diarrhoeagenic pathogens from genera such as *Salmonella*, *Yersinia*, *Escherichia* and *Shigella* are a significant health burden worldwide, with diarrhoea causing over 1.6 million global deaths per year as recently as 2016 [2]. Large-scale studies of children in developing countries confirmed the contribution of enteropathogenic *Escherichia coli* (EPEC) and *Shigella* to severe diarrhoea and mortality [3,4]. *Shigella* can be considered a form of enteroinvasive *E. coli* (EIEC); the two pathovars have seemingly arisen through convergent evolution of multiple lineages after the acquisition of horizontally acquired virulence genes [5], and so present both medical and diagnostic challenges [6,7,8]. Antibiotic-resistant *Shigella* remains a significant problem in the western world and has even found a niche as a sexually transmitted infection [9,10]. Studies in Europe have highlighted the considerable economic cost of *Yersinia* infection, which may be dramatically underreported [11,12]. Similarly, Shiga-toxigenic *E. coli* (STEC), responsible for haemolytic uremic syndrome, was previously estimated to result in over 175,000 infections annually in the USA [13] and caused a large, multistate outbreak in 2018 [14]. Note that, in this review, we will use the term STEC to refer to the broader class of Shiga-toxigenic strains (including those that do not encode the locus of enterocyte effacement (LEE)) and reserve the term enterohemorrhagic *E. coli* (EHEC) to refer to LEE-encoding strains such as O157:H7.

The virulence factors employed by diarrhoeagenic pathogens range from single-protein exotoxins to complex, macromolecular assemblies anchored within the bacterial cell wall. Such virulence factors include flagella, fimbriae (pili) and secretion systems (SSs).

## 2. Secretion Systems in Enterobacterial Pathogenesis

In order to interact with the external environment, bacteria must secrete proteins to the cell surface or the external medium. In Gram-negative and other diderm bacteria such as *Mycobacterium* spp., the presence of an outer membrane poses an additional obstacle for secreting macromolecules to the outside of the cell. Nevertheless, diderm bacteria have evolved several distinct secretion pathways to translocate proteins across the cell envelope. Currently, there are nine “official” SSs designated with a number (type 1 secretion, type 2 secretion, etc.) as well as the chaperone–usher pathway for constructing pili [15]. In addition, some pathways have been described that have not (yet) been accepted into the canon [16,17,18]. Enterobacteria use many of these systems to export toxins or even to inject them directly into the host cell cytoplasm. Such injected toxins are generally referred to as “effectors”, which are secreted by type 3 SSs (T3SSs), T4SSs and T6SSs. In addition, enterobacteria rely on adhesins for host cell contact in order to inject effectors or to efficiently deliver extracellular toxins to target cells. Adhesins are generally secreted by the T5SS or assembled by the chaperone–usher pathway, or the T2SS for type IV pilus assembly [19]. In enterobacteria, the T5SS and T3SS often act in concert to promote effector injection, typically leading to rearrangements of the cytoskeleton. The maintenance of the host cytoskeleton plays a key role in the preservation of the cellular structure, vesicular transport, and the highly conserved regulation of cellular permeability. However, infections with pathogens encoding the T3SS often interfere with these essential structures [20]. Here, we review how these two secretion systems, the T3SS and T5SS, synergize to allow bacteria to invade host cells, escape phagocytosis, confer intracellular motility or change cellular morphology.

### 2.1. Type 3 Secretion Systems

#### 2.1.1. Overview

The T3SS or “injectisome” is a complex structure that likely shares an evolutionary origin with flagella (reviewed in [21]). It functions as a molecular syringe—an ATP and proton motive force-dependent unfolding and secretion mechanism [22] that transports effector proteins from a bacterial cell’s cytoplasm directly into another target cell, eukaryotic or otherwise. Both invasive and non-invasive enteropathogens carry structurally homologous T3SSs, the primary difference between them being the effector proteins that are delivered into the host. In diarrhoeagenic *E. coli*, *Shigella* and *Yersinia*, the target cell is often part of the human intestinal epithelium, although T3SS also have an important anti-phagocytic function that will be explored below.

Structurally, the T3SS spans both the inner and outer membrane of the Gram-negative cell (Figure 1) and is composed of dozens of distinct proteins that have close homologues throughout the Enterobacterales. The main structural units of the T3SS are often encoded in pathogenicity islands and, due to the % GC content of the genes compared to genomic DNA and phylogenetic comparisons between T3SS variants, it seems likely that such systems are primarily horizontally acquired [23]. Indeed, in *Yersinia enterocolitica* and *Y. pseudotuberculosis*, as well as in EIEC and all four *Shigella* species, the T3SS required for host epithelium invasion is encoded within a pathogenicity island carried on a virulence plasmid. T3SS-encoding plasmids vary in size from approximately 70 kb (*Yersinia* pYV; [24]) to over 290 kb (EIEC pINV; [5]). *Shigella* pINV, at approximately 220 kb, is more similar to pINV from EIEC in terms of gross composition and function than it is to pYV [25].

Chromosomal T3SSs often serve a different function to their plasmid-borne counterparts. In EPEC and EHEC strains, a T3SS encoded within the chromosomal LEE provides the organism with a mechanism of adherence that does not typically lead to host cell invasion (reviewed in [26]). Instead, the LEE T3SS injects an effector protein, translocated intimin receptor (Tir), into the host epithelial cell in order to initiate actin pedestal formation. The role of Tir is discussed in Section 3 below and demonstrates one way in which T3SS and T5SS function together. The minimal “core” T3SS locus is approximately 30 kb [27].

#### 2.1.2. Architecture and Assembly of the T3SS

The T3SS (Figure 1) contains three major components: a basal body that is anchored within the inner and outer membranes of the bacterial cell; a hollow needle passing through the basal body to beyond the cell envelope; and a needle tip complex referred to as the translocon. The basal body contains an ATP-dependent export apparatus [30] that recruits and unfolds effector proteins in order for them to pass through the needle. The translocon acts both as a regulatory “plug” for the needle and as a pore-forming complex, inserting a pore in the target cell membrane and initiating secretion from the needle in response. Though it was the subject of debate for some time, in an elegant experiment using a “blocked” T3SS needle, Dohlich and coworkers demonstrated that effector proteins do indeed travel through the central needle of the T3SS in order to reach target cells [32]. Similarly, Radics et al. used cryo-electron microscopy to visualize the *Salmonella* effector SptP as an unfolded mass within the needle [33]. Hence, the needle/translocon does not simply act as a signal transduction mechanism for other types of protein export [34]. However, in recent years, it has become apparent that the T3SS can also mediate the transport of some externally added effectors into host cells [35,36].

Assembly of a T3SS and subsequent protein secretion appears to be a strictly hierarchical process with homology across multiple species (reviewed in [30]). The basal body is the first major part of the T3SS to be constructed, likely upon a series of scaffold protein rings in both the inner and outer membrane. Associated with this structure is a cytoplasmic ring (C-ring) that forms a complex with an essential ATPase and accessory proteins.

After the secretion of needle subunits through the basal body and their assembly into a helical structure [37], the translocon proteins are likewise secreted to the needle tip. The proteins SctE (for secretion and cellular translocation protein E), SctB and SctA (*Shigella* invasion plasmid antigen (Ipa) B, IpaC and IpaD; *Yersinia* outer protein (Yop) B, YopD and low calcium respeons protein (Lcr) V; *E. coli* secreted protein (Esp) D, EspB and no homolog, respectively) form the translocon [31,38,39,40]. Whilst SctA controls secretion whilst at the needle tip (in some cases together with SctE [39,41]), SctE and SctB associate upon activation of secretion, forming a pore in the target cell membrane [42]. Upon host cell contact, the translocon transduces a signal back into the cytoplasm; sometimes this signal is due to host contact-dependent conformational changes in proteins (e.g., IpaD [42]) from the tip downwards, whereas in other T3SSs, the process is chemical in nature, such as a change in pH [43]. A so-called “gatekeeper” protein, localized in the cytoplasm, SctW, serves to prevent effector protein passage through the export apparatus and is typically displaced by the mechanical or chemical changes after host cell contact [43]. For example, the gatekeeper MxiC’s release from the *Shigella* T3SS structure allows for secretion of bona fide effector proteins [44]. Similarly, during regulation of the EPEC LEE T3SS, the MxiC-like protein SepL binds specifically to Tir and delays its secretion until the translocon has correctly formed [45]. In *Yersinia*, an extra layer of regulation exists in the form of a YopN-dependent “calcium block” that represses expression and secretion of effector proteins in either the presence of calcium or the absence of target cell contact [46]. The structure of the YopN–TyeA complex involved in this process is somewhat reminiscent of MxiC [47] and SepL [45].

#### 2.1.3. Effector Proteins

The substrates secreted into host cells through the T3SS, known as effector proteins, are typically encoded both in the pathogenicity island (alongside the T3SS structural genes) as well as elsewhere on the relevant virulence plasmids and chromosomal DNA [27,30,48,49]. Whilst there is significant homology evident in the structural T3SS proteins from different species, their effectors are much more species-restricted (Table 1). Effector proteins are divided into “early” and “late” effectors, the former category including proteins already discussed in *Shigella*, such as IpaB and IpaC, that regulate secretion and pore formation. Other early *Shigella* effectors include proteins such as IpaA [50] that interact specifically with host proteins to trigger changes in the host cytoskeleton, and IcsB, which acts as a chaperone to protect the invading microbe from autophagy induced by the autotransporter Intercellular spread protein A (IcsA) [51]. There is also evidence that the translocon protein IpaC can itself cause host cytoskeletal modification [52]. Its EPEC homologue, EspB, similarly has a multifunction role as a secreted immune suppressor [53,54] as well as translocon component, and is discussed in more detail later in this review.

Exploring the functions of all T3SS effector molecules across the various diarrhoeagenic pathogens is beyond the scope of this review, though this has been covered in excellent detail elsewhere [28,55,56,57,58]. Typically, effectors act as “cytoskeletoxins” that interact with the host cell cytoskeleton in order to cause macropinocytosis of the pathogen, rearrangements of the actin cortex, or to disable the host immune system. Key effectors are described for individual species in their respective sections, where appropriate. 

### 2.2. Type 5 Secretion Systems

T5SSs are the simplest and most widespread secretion systems among Gram-negative bacteria [99]. T5SSs are often referred to as autotransporters, due to the original view that these constitute self-contained secretion systems [100]. However, work during the past two decades has demonstrated that T5SSs are dependent on several constitutive machineries of the bacterial cell, including the Sec translocon, periplasmic chaperones and the β-barrel assembly machinery in the outer membrane [101,102]. There are currently five recognized subclasses in the T5SS scheme. Three of these are relevant for this review: T5aSSs or classical autotransporters, T5cSSs or trimeric autotransporter adhesins (TAAs), and T5eSSs or inverse autotransporters (Figure 2). These three share the same basic architecture, with an outer membrane-embedded β-barrel translocator and an extracellular region or passenger. The passenger harbours the specific function of the autotransporter, whereas the β-barrel domain is required for passenger secretion. In classical autotransporters, the passenger is at the N-terminus of the protein and is in many cases proteolytically cleaved from the β-barrel [103]. By contrast, inverse autotransporters have the opposite topology to classical autotransporters, and consequently the passenger is at the C-terminus of the protein [104]. TAAs have a topology similar to classical autotransporters, but these are obligate homotrimeric proteins where both the N-terminal passenger and the β-barrel domain are formed by three identical polypeptide chains [105,106]. Classical autotransporters may have a number of functions, ranging from adhesion and autoaggregation to enzymatic activity and functioning as toxins. However, all characterized inverse autotransporters and TAAs have adhesive functions, but may also be involved in immune evasion or host cell invasion [103]. T5SSs found in enterobacterial pathogens are summarized in Table 2. 

## 3. Staying out: The Attaching and Effacing Pathogens

The attaching and effacing (A/E) pathogens, including EPEC and EHEC, are primarily extracellular bacteria characterized by binding to the luminal surface of enterocytes, which leads to the resorption of microvilli (effacing of enterocytes) and the formation of a protrusion from the cell, called a pedestal, that acts as a platform for these bacteria (Figure 3). Microvillus effacement and pedestal formation are driven by rearrangement of the actin cortex. These phenomena are dependent on a horizontally acquired pathogenicity island, the LEE, which encodes a T3SS, effector proteins, transcriptional regulators, and an adhesin, intimin [126]. Pedestal formation and intimate contact are necessary for virulence of A/E pathogens, and the LEE is an essential virulence factor. While the function of pedestal formation remains somewhat mysterious, this presumably prevents dissociation of the bacteria from the epithelium and promotes the changes needed for diarrhoeagenesis [127]. In addition, recent evidence points to pedestal formation protecting the bacteria from phagocytosis [128].

The LEE can vary in size and composition. The ~35 kb LEE from the EPEC strain E2349/69 can be considered to represent a “core” LEE, whereas other A/E strains contain flanking regions that can extend the pathogenicity island to over 100 kb in length [126]. The flanking regions often contain insertion elements, prophage sequences, or additional effector proteins. In some strains, the 3′ flanking regions contain an element encoding the non-LEE effectors Ent, NleA, NleB and a large toxin, lymphostatin, which has been implicated in haemolytic uremic syndrome [78,126,129]. 

The LEE is subject to complex genetic regulation. The core LEE itself encodes three regulators, Ler, GrlA and GrlR [130]. Ler is the master regulator of the other LEE genes. At temperatures below 37 °C, the histone-like nucleoid structuring protein H-NS represses *ler* and other LEE genes; at 37 °C, *ler* is expressed and leads to the expression of other LEE-encoded factors. However, many other environmental cues regulate *ler* expression, including quorum sensing, oxygen levels, biotin concentration, intracellular iron, host hormones, metabolites such as butyrate, fucose, and phosphorylated fructose species, and attachment to host cells [131]. Furthermore, the LEE is internally regulated by the interplay of GrlA, a positive regulator, and GrlR, a negative regulator, which fine tunes the expression of LEE components [132,133]. Thus, the regulatory network controlling LEE expression must integrate a variety of signals to finely tune the expression levels of the LEE-encoded factors [130,131].

### 3.1. The Esc-Esp T3SS

The Esc T3SS (also called ETT1) of EPEC and EHEC is encoded by the LEE. This is a major virulence factor of both organisms and is responsible for translocating toxic effectors, generally called *Escherichia coli* secreted proteins (Esps). The main targets of the Esc T3SS are the intestinal epithelial cells, but Esc can also target immune cells. The effectors translocated by the Esc T3SS of EPEC and EHEC lead to rearrangement of the cytoskeleton, subvert innate immune responses, compromise the epithelial barrier function and modulate cell survival [53]. Seven effectors are encoded by the LEE (EspB, EspF, EspG, EspH, EspZ, Map and Tir), but in addition, several non-LEE effector genes located elsewhere in the genome have been identified. Typical EPEC strains encode ~20 effectors, whereas some EHEC strain encode close to 40 [26]. Therefore, only the “core” Esp effectors and a couple of noteworthy non-LEE effectors will be considered here (Table 1).

The LEE-encoded Esc substrates include EspA, EspB and EspD (Table 1). These are not effector proteins as such, but rather play a structural role. EspA polymerizes to form the extracellular filament connecting the needle to the host cells, whereas EspB and EspD form the translocon pore in the host cell membrane (Figure 1) [26]. In addition, EspB has a true effector function in host cells in preventing phagocytosis. EspB interacts with myosin, preventing the formation of pseudopods and phagosome closure [53]. Other effectors that influence phagocytosis include EspF, EspH and EspJ (Table 1). EspF inhibits phagocytosis by interfering with phosphoinositide 3-kinase signalling and subsequent F-actin rearrangements [53]. EspF is a multifunctional protein, and in addition to preventing phagocytosis, also targets tight junctions in epithelial cells, contributes to microvillus effacement and can induce apoptosis [60]. EspJ prevents phagocytosis of opsonized cells by macrophages through ADP-ribosylating several cytoplasmic host kinases [62]. EspH prevents activation of the small GTPases of the Rho family by binding to nucleotide exchange factors [53]. Map, EspM and EspT are all guanine nucleotide exchange factors that modulate cytoskeletal function through small GTPases (Table 1); Map activates Cdc42 and promotes the formation of filopodia, EspM targets RhoA to increase stress fiber production, and EspT activates both Cdc42 and Rac1 leading to membrane ruffling and the formation of filopodia [134]. The final LEE-encoded effector, EspZ, regulates the translocation of other effectors and thereby modulates their intracellular activity [63].

The most abundantly secreted effector is the translocated intimin receptor, Tir. This is also the central effector protein for virulence and binding of Tir by intimin is sufficient to promote actin cytoskeleton rearrangements leading to pedestal formation in EPEC [135]. However, it should be noted that in vivo, other effectors including non-LEE effectors are needed to form full A/E lesions by EPEC [48]. In EHEC, a second non-LEE effector, EspF_U_ (formerly TccP), interacts with Tir to mediate pedestal formation [61]. Pedestal formation is discussed in more detail in Section 3.3.

Interestingly, the Esc T3SS does not only translocate classical effector proteins. In EPEC, the T5a autotransporter EspC is also transported into host cells in a T3SS-dependent manner [136]. EspC is a serine protease that is first secreted into the extracellular medium by the T5SS, after which the passenger is autoproteolytically cleaved from the β-barrel domain [112]. Outside the bacterium, EspC regulates T3SS function externally by cleaving the translocon protein EspD and the filament protein EspA [137]. EspC only targets the EspA–EspD complex that is secreted upon host cell contact; however, polymerized EspA in the injectisome filament is not degraded by EspC. The result of this degradation is regulation of the number of T3SS pores formed on the host cell and a subsequent reduction in cytotoxicity of the pores. EspC thus acts as a negative regulator of the T3SS and prevents premature cell death of epithelial cells due to T3SS cytotoxicity [137]. The related protein EspP of EHEC has a similar function to EspC and can cleave EspA, EspB and EspD to negatively regulate the Esc T3SS [138]. 

EspC apparently has some other extracellular functions. Like the EHEC protein EspP, the EspC passenger can form multiprotein filamentous structures called “ropes”, which have cytopathic and adhesive properties and can act a substrate for biofilm formation by the bacteria [139]. Further, EspC is able to degrade hemoglobin and bind to hemin, which suggests that EspC may play a role in iron acquisition by EPEC [140].

More surprising are the intracellular effects of EspC, where it causes changes in the actin cytoskeleton, cell rounding and detachment [112]. It does this by targeting fodrin, a protein that anchors the actin cortex to the plasma membrane, and two focal adhesion proteins, paxillin and focal adhesion kinase (FAK) [141]. In addition, procaspase-3 is activated, leading to apoptosis [142]. To enter host cells, the EspC passenger binds to EspA on the outside of the injectisome and then interacts with EspB and EspD to enter the host cell [36]. The relationship between the T5SS EspC and the Esc T3SS demonstrates again how the two secretion systems often act together to promote virulence. To our knowledge, there are no published studies or hypotheses for how the dual (extra- and intracellular) functions of EspC might have evolved, but its several functions are in keeping with the observations that many autotransporters are multifunctional proteins. 

### 3.2. Intimin

Intimin is an adhesin and an essential virulence factor of A/E pathogens. It is also the prototype of the T5eSS [104,143]. Intimin consists of a 12-stranded β-barrel domain followed by a passenger formed by four immunoglobulin(Ig)-like domains capped by a lectin-like domain at the C-terminus [144,145]. The receptor binding region is formed by the terminal Ig-like domain and the lectin-like domain [144]. In addition, intimin has a short N-terminal periplasmic region including a peptidoglycan-binding LysM minidomain [146]. The two C-terminal domains form a superdomain that interacts with its receptor, Tir. Both proteins function as dimers: Tir recruits two intimins but, as shown by the crystal structure of the complex, these are most likely from two separate intimin dimers [144,147]. This would lead to an intimin–Tir network at the attachment interface, which would cluster intracellular signalling molecules, leading to downstream effects promoting pedestal formation (see 3.3. below) [146,147].

There are 18 different subtypes of *eae*, the gene for intimin, with at least 27 alleles [115]. The most clinically important types are intimin-α, associated with typical EPEC strains, intimin-β, found in some EPEC and EHEC strains and *Citrobacter rodentium*, and intimin-γ, associated with EHEC O157:H7. The different intimin isoforms have been implicated in the tissue tropism of the producer bacteria. For example, EHEC usually colonizes the epithelium covering ileal Peyer’s patches, the lymphoid follicles of the intestinal tract. In contrast, EPEC colonizes the epithelium of the small intestine. Introducing the *eae* gene encoding intimin-γ into a Δ*eae* EPEC strain switched the tropism to Peyer’s patches [148]. The intimin subtype may also play a role in host specificity: intimin-γ cannot complement the lack of intimin- β in a murine infection model with *C. rodentium*, and conversely intimin-β cannot restore normal colonisation patterns to a Δ*eae* EHEC strain in a porcine organ culture model [149]. However, more recent work, where intimin subtypes were expressed from the chromosomal locus rather than a plasmid, did not show similar changes in tissue tropism [150]. Thus, the role of intimin subtypes in the tissue tropism of different pathogens still requires clarification.

### 3.3. Pedestal Formation

After translocation into the host cell, Tir integrates into the host plasma membrane to act as the receptor for intimin [93]. Tir is a bitopic transmembrane protein with both the C- and N-termini located in the host cell cytoplasm [151]. Tir also forms homodimers, and the dimerisation is mediated by the extracellular intimin-binding domain [144]. Dimerisation of Tir is important for downstream effects, as this leads to formation of an intimin–Tir network needed for inducing actin rearrangements and pedestal formation [147,150]. In EPEC infections, a central step leading to actin cytoskeletal remodelling is Tir phosphorylation at Y474 [152]. This is mediated by various kinases and is dependent on intimin mediated-Tir clustering [153,154,155,156]. Phosphorylated Y474 binds the adaptor protein Nck [157]. This leads to recruitment of neural Wiskott–Aldrich syndrome protein (N-WASP) and ultimately actin polymerisation via the Arp2/3 complex. A recent study demonstrated a role for the formin mDia1 in pedestal assembly [158] 

By contrast, EHEC uses a different mechanism for pedestal formation. EHEC Tir is not dependent on tyrosine phosphorylation; in fact, it lacks an equivalent to Y474 [159]. It is also independent of the Nck pathway. Rather, actin assembly for pedestal formation requires an EHEC-specific, non-LEE-encoded effector, EspF_U_, which binds to Tir in the host cytoplasm [61]. EspF_U_ recruits two host proteins, IRSp53 and IRTSK, to form a complex that activates N-WASP and Arp2/3 [160,161]. As for EPEC, this leads to actin polymerisation at the site of intimate contact and pedestal formation. The difference in the way actin polymerisation takes place in EHEC and EPEC infections explains why EHEC Tir cannot complement the Tir from EPEC. The EspF_U_ pathway has some differences in outcomes compared with the Tir-Nck pathway used by EPEC: firstly, the EspF_U_ pathway promotes a greater degree of adhesion and more colonisation of the epithelium than the Tir-Nck pathway [162]. Secondly, the EspF_U_ pathway promotes more efficient cell-to-cell spreading. A/E pathogens can use a type of actin-based motility (ABM) called surfing motility, where the bacteria together with the entire actin pedestal glide along the surface of the cell. This allows spreading of the bacteria over the apical surface of the epithelium and the formation of “microcolonies”; the EspF_U_ pathway is more efficient in this and leads to faster and larger microcolony formation [128].

The intimin–Tir interaction is needed for intimate attachment and pedestal formation. However, if binding of intimin to Tir is needed for efficient effector translocation, this creates a chicken-and-egg problem: how can intimin mediate translocation of Tir if Tir is not already present on the host cell surface? A possible explanation is that intimin binds to endogenous receptors on enterocytes, which would also explain the different tissue tropism seen between different intimin subtypes [148,149]. Intimin has been suggested to bind to β _1_ integrins and cell surface-localized nucleolin [163,164,165]; these interactions could aid in docking the Esc T3SS to the host cell surface to inject effectors. However, the interactions with host proteins remain poorly characterized and the binding to β _1_ integrins has been contested [166]. In addition, intimin interactions have been widely investigated using “prime-challenge” experiments, where cell cultures are first infected with a Δ*eae* EPEC strain, leading to Tir translocation into the cells [135,150,167]. The bacteria are then removed before adding intimin-expressing laboratory *E. coli* strains that bind to the Tir-containing host cells. This experimental set-up strongly suggests adhesins other than Tir are needed for the initial binding and initiation of effector secretion. In EPEC, candidates for initial adhesion include bundle-forming pili [168] and *E. coli* common pili [169], as well as several autotransporters (Table 2). It also appears that the T3SS itself can mediate a low level of adhesion [168,170]. In EHEC, long polar fimbriae and other pili and several type 5a and 5c adhesins (Table 2) might contribute to initial adhesion and T3SS docking [115].

## 4. Going in: EIEC and *Shigella*

Both *Shigella* and EIEC are characterized by their ability to invade the human intestinal epithelium followed by intracellular replication and dissemination. As depicted in Figure 4, the highly adapted organism firstly invades through the human microfold (M) cells located in the epithelium covering Peyer’s patches, the lymphoid follicles of the intestine [171]. The subsequent release to the basolateral side of the epithelium results in uptake of the bacteria by the host macrophage. Here, the avoidance of phagocytosis is achieved via rapidly induced apoptosis. This allows the bacteria to invade the epithelium from the basolateral side, where they can spread from cell to cell resulting in inflammatory destruction. *Shigella* are a group of bacteria incredibly adapted to life inside of the human host having lost a substantial number of genes vital for survival in the environment yet unnecessary inside the host. The highly adapted modifications in metabolic and pathogenic profiles of *Shigella* are not yet as advanced in EIEC. EIEC, being metabolically closer to *E. coli*, has not experienced the loss of anti-virulence genes and adaptive mutational events limited to *Shigella* [25]. The metabolic differences between the two species allow differentiation of the two pathogens, even though 16S RNA-based differentiation is not possible. For example, the utilisation of lactose on MacConkey medium is a classic way to separate the two species [172].

### 4.1. The Invasion Process

It is widely acknowledged that the acquisition of the large virulence plasmid, pINV, in both *Shigella* and EIEC was fundamental to establishing their virulence, enabling the bacteria to invade and spread through the human intestinal epithelium resulting in the inflammatory destruction referred to as bacillary dysentery [173]. Essential for this process, pINV produces a T3SS encoded by the conserved *mxi-spa* locus, allowing the pathogens to establish a safe and nutrient-rich environmental niche inside the host [174]. Not only does the T3SS control invasion into the host intestinal epithelium but further controls the intra- and intercellular spread, macrophage cell death and host inflammatory responses. The pINV-encoded virulence genes required for infection are under tight control via the primary transcriptional regulator VirF, with their expression governed by various environmental stimuli within the host.

The *Shigella* invasion process is primarily regulated by the temperature of the human body. The transcriptional regulator VirF is optimally expressed at 37 °C, resulting in the activation of *virB* and *icsA* expression [175]. The *Shigella* T3SS is also precisely regulated in response to varying levels of oxygen throughout the gastrointestinal tract. In anaerobic conditions present in the gastrointestinal tract lumen, Ipa effector protein secretion is significantly reduced and the T3SS needle is extended, although is not yet effective for secretion. This regulation is mediated by the FNR transcriptional regulator, which is active in low oxygen conditions and is responsible for the repression of *spa32* (*sctP*; regulating needle length) and *spa33* (*sctO*; the C-ring) [176,177]. The area adjacent to the mucosa is, however, sufficiently oxygenated to prevent FNR activity and trigger T3SS activation for effective invasion. This complex regulation allows the accumulation of virulence proteins in anaerobic conditions, ready for use once the bacteria encounter the epithelium [176,178].

Following T3SS assembly at 37 °C, contact with M cells [171] is required to facilitate the translocation of effector proteins into the host cell cytosol through the T3SS needle [44]. The host filament vimentin, whilst dispensable for pore formation itself, triggers docking of the T3SS to the pore [179], a process that is enabled by a conformational change in IpaC [180]. The *Shigella* proteins IpaB, IpaC and IpaD form the translocon [38,39,40], and IpaD and MxiC regulate the secretion of effectors [41,44]. The secreted effectors (Table 1) trigger cytoskeletal remodelling, consequently resulting in the formation of transient actin-rich membrane ruffles which engulf the infecting bacterium within a vacuole [42]. Following transcytosis, *Shigella*/EIEC are released from the M cell where they encounter resident macrophages [55]. These pathogens are able to survive the degrading effects of macrophages through the rapid induction of apoptosis prior to their release into the basolateral surface of the epithelium, from where they invade the neighbouring epithelial cells via a macropinocytic process [174,181]. At the basolateral surface, glucosylation of lipopolysaccharides allows the T3SS to contact the epithelium in addition to increased immune evasion. It is thought that IpaB, localized at the needle tip, binds to C44 lipid microdomains to uphold contact to the epithelial cell [28,182]. Faherty et al. further noted that *ospE1/ospE2* genes were induced in the presence of bile. The OspE1/OspE2 effectors secreted by the T3SS were shown to remain localized on the bacterial outer membrane where they may act as adhesins in response to bile salts [87]. At this stage of infection, proinflammatory cytokines interleukin (IL)-1β and IL-18 are released, characteristic of the large inflammatory response associated with shigellosis [55,183].

### 4.2. Actin-Based Motility

*Shigella* and EIEC are typically non-flagellated. Once in the epithelial cell cytoplasm, the bacterium begins recruiting host actin to form “actin tails” used to propel the bacteria, allowing intracellular movement and dissemination from cell to cell. This results in rapid lateral spread through the epithelial layer without the need for re-entry through the basolateral pocket [184,185]. This method of infection allows the bacteria to reside and replicate within the host cells, minimising exposure to the immune response. This ABM is highly dependent upon the surface-exposed type 5a autotransporter IcsA, also known as VirG. IcsA is regulated by VirF and is responsible for the recruitment and polymerisation of the host actin at one pole of the bacterial cell [186,187]. The N-terminus of the passenger interacts with host proteins such as vinculin and N-WASP, leading to the subsequent actin polymerisation at one pole of the bacterium [188,189]. 

In addition to ABM, the multifunctional role of IcsA as an adhesin has also been investigated. Following the knockout of T3SS translocon proteins IpaD and IpaB in *S. flexneri,* polar adhesion to host cells was observed in an IcsA-dependent manner [117]. The expression of IcsA in *E. coli* further supported this hypothesis as the protein was found to promote adherence to host cells. However, detectable adherence was not identified unless exposed to environmental stimuli, such as the bile salt deoxycholate [190], or via T3SS activation [117]. Qin et al. went on to identify the specific binding region within the passenger domain responsible for adhesion [191]. Interestingly, an *icsA* adhesion-defective mutant obtained via insertions into the passenger resulted in an attenuated infection phenotype using the Sereny test [117]. In addition to its roles in ABM and host cell adhesion, IcsA also plays a key role in *S. flexneri* biofilm formation in the presence of deoxycholate by promoting cell-to-cell contact and aggregative bacterial growth [192]. This multifunctional protein therefore plays a key role in *Shigella* pathogenesis specifically promoting ABM, host cell adhesion and biofilm formation under varying environmental niches. IcsA is therefore both necessary and sufficient for host cell contact enhancing pathogenesis. Although to our knowledge this has not been studied, it is possible that IcsA is also needed for docking the T3SS to the host cell surface, similarly to YadA in the enteropathogenic *Yersiniae* (see Section 5.2).

### 4.3. EIEC vs. Shigella

Despite the remarkable similarities between EIEC and *Shigella* invasion, it has been well documented that EIEC infection requires a higher infectious dose, resulting in a much milder disease compared to *Shigella* [25]. The main differences centre around macrophage escape and intracellular dissemination [193,194]. Induced apoptosis resulting in macrophage escape is a key step in intestinal colonisation for both *Shigella* and EIEC. However, comparative studies have demonstrated EIEC’s reduced efficiency of murine J774 macrophage escape and cell-to-cell spread, as well as the lowered effectiveness of cell killing during the first 4 h post infection when compared to *Shigella* [194]. During earlier stages of *Shigella* infection, Moreno and co-workers observed that the keratoconjunctivitis model gave more severe results, with *Shigella* inducing a greater proinflammatory response compared EIEC infection [193]. Whilst EIEC and *Shigella* share the same virulence factors and mode of invasion [195], despite the clear differences in disease development, investigation turned to the relationship between EIEC virulence gene expression profiles and the reduced ability to cause disease.

During invasion, *ipaABCD* expression is reduced in EIEC compared to *S. flexneri.* More specifically, *ipaC* showed the greatest diminished expression during EIEC intracellular infection despite its key role in phagosome escape [196,197] and epithelial invasion [193,198] thus supporting the diminished macrophage death and cell-to-cell spread previously mentioned. Interestingly, this reduced expression is not true for all EIEC invasion-associated virulence genes. Moreno et al. [193] documented higher levels of *virF* expression in EIEC compared to *Shigella* intracellularly during dissemination. This might be attributed to the translation of *virF* mRNA into two separate proteins of 30 kDa and 21 kDa [199]. The shorter protein negatively autoregulates *virF* expression, modulating intracellular VirF levels. The large variations in disease severity despite the similar mode of infection highlights the importance of the efficiency of virulence gene transcription and regulation. These variabilities in gene regulation and response to environmental stimuli may underpin the resulting differences in disease outcomes.

Overall, the acquisition of the T3SS was essential for both *Shigella* and EIEC to invade, disseminate and survive within the host during infection. Their unique mode of infection demonstrates the highly conserved regulation of essential virulence genes involved in infection and its ability to cause disease. The complex regulation involved in T3SS expression is precisely adapted to the human host, resulting in extremely efficient invasion and spread leading to severe disease. Both the T3SS and the autotransporter IcsA result in changes to the actin cytoskeleton, with IcsA also playing a key role as an adhesin enhancing pathogenesis. *Shigella* is highly adapted for optimal invasion within the human host, especially when compared to EIEC, portraying the importance of virulence gene expression efficiencies.

## 5. Maybe Going in? The Enteropathogenic *Yersiniae*

The enteropathogenic *Yersiniae*, *Y. enterocolitica* and *Y. pseudotuberculosis*, have a very similar lifestyle and mode of infection. Both organisms are transmitted by the faecal–oral route. Once they reach the terminal ilium, the bacteria invade M cells [200]. The bacteria are transcytosed through the M cells and released into the Peyer’s patch, where they establish a primarily extracellular infection. From the Peyer’s patches, the enteropathogenic *Yersiniae* disseminate to the mesenteric lymph nodes to cause a usually self-limiting infection. In rare cases, the bacteria can spread to other organs, particularly the liver and the spleen, to cause a systemic infection. There is evidence for direct dissemination from the intestinal lumen to the liver, possibly via the hepatic portal vein [201,202]. This can happen particularly in cases of iron overload in the bloodstream, e.g., due to genetic factors [203]. In addition, a pool of replicating bacteria remains in the intestinal lumen, which is presumably responsible for the diarrhoeal component of the infection [201].

Pathogenesis of enteric yersiniosis is dependent on three major virulence factors. These are the autotransporter adhesins invasin (InvA) and YadA and the plasmid-encoded T3SS, Ysc, and its effectors, the *Yersinia* outer proteins or Yops [203]. These three virulence factors, InvA, YadA and Ysc-Yops, all elicit cytoskeletal changes in host cells and act in concert to establish an extracellular infection in lymphatic tissues. The paradigm for the classical *Yersinia* infection route has been that InvA is expressed first and mediates invasion of the intestinal epithelium. Once within the submucosa, YadA and Ysc are expressed and together promote an extracellular infection in the lymphatic tissues by targeting leukocytes and preventing phagocytosis of the bacteria (Figure 5A). The latter two factors are essential for virulence in *Y. enterocolitica*, whereas lack of InvA delays the onset of infection [204,205]. In contrast, only Ysc is essential for *Y. pseudotuberculosis* pathogenicity [206]. Interestingly, in the plague bacillus *Y. pestis*, both *invA* and *yadA* are inactivated though Ysc is still required for infection [207]. The lack of the adhesins probably reflects the altered infection route of *Y. pestis*: this organism is transmitted subcutaneously by flea bites, or by aerosols in the case of pneumonic plague, not through the intestinal tract. 

Although the *Yersiniae* have been considered mainly extracellular pathogens, there is mounting evidence that they can invade host cells other than M cells, and that they can survive and even replicate within neutrophils and macrophages [208,209,210,211,212]. Recently, uptake of *Y. enterocolitica* by dendritic cells in intestinal tissues was demonstrated [202]. The same study showed that *Y. enterocolitica* can disseminate to the spleen independently of Peyer’s patches and mesenteric lymph nodes, and that this was dependent on mononuclear phagocytes. *Y. pseudotuberculosis* and *Y. pestis* encode a type 5a autotransporter, YapV, which is similar to IcsA and can interact with N-WASP [213]. This suggests that these bacteria are capable of ABM, again supporting the view of an occasional intracellular lifestyle. Thus, the paradigm of an obligate extracellular lifestyle once *Yersiniae* have crossed the intestinal epithelium should be revised to consider the facultative intracellular nature of these pathogens. 

### 5.1. Invasin

InvA is expressed at environmental temperatures (26 °C and below) as well as at higher temperatures under acidic conditions [214]. The bacteria are therefore thought to be primed with InvA on the surface when ingested and after transit through the gastrointestinal tract. Recently, InvA was also showed to be upregulated during persistent infections [215]. 

InvA is a chromosomally encoded inverse autotransporter similar in structure to intimin [145,216]. However, unlike intimin, InvA binds directly to cellular receptors, the β_1_ integrins [217]. These are not normally present on the apical surface of enterocytes; however, M cells do express β _1_ integrins on their apical surface [218,219]. The enteropathogenic *Yersiniae* bind to these receptors via InvA, which leads to internalisation of the bacteria by a zipper-like mechanism (Figure 5B). This is a consequence of integrin clustering, which leads to intracellular signalling events and the formation of focal adhesions and actin rearrangements [118]. The *Y. pseudotuberculosis* InvA protein is the more potent mediator of invasion due to the presence of a self-interaction domain (D2) in the passenger lacking in the *Y. enterocolitica* orthologue [220,221]. Integrin clustering leads to phosphorylation of kinases such as FAK and Src, phosphorylation of the adapter protein Cas and recruitment of Crk, as well as activation of the GTPase Rac1 [222,223,224]. This leads to the cytoskeletal arrangements mediated by WASP family members and the Arp2/3 complex resulting in the engulfment of the bacteria by the zipper mechanism [223,225].

### 5.2. YadA

In contrast to *invA*, the expression of *yadA* is upregulated at 37 °C, the mammalian body temperature [226]. YadA is the prototypical TAA, consisting of a globular head at the N-terminus, a coiled coil stalk and a C-terminal β-barrel domain. It is encoded on the *Yersinia* virulence plasmid pYV [205]. YadA is a multifunctional protein that promotes adhesion to cells and extracellular matrix (ECM) components, host cell invasion, serum resistance by binding to complement-regulatory factors, as well as autoaggregation and biofilm formation [122,143]. YadA thus likely contributes to microcolony formation within the infected tissues by protecting the bacteria from serum and mediating biofilm-like structures that are resistant to phagocytosis [227,228]. In addition, YadA acts as a scaffold for docking the Ysc T3SS apparatus onto host cells [229,230]. The length of the YadA stalk and the Ysc injectisome needle are correlated; genetically increasing the length of YadA or shortening the injectisome led to inefficient Yop delivery [231]. Both YadA and InvA can provide the docking function for Ysc, but YadA usually has a much more prominent role for Yop delivery in vivo, though the relative contributions of the adhesins to Yop delivery are strain-dependent [229,230,232]. The model for the interaction is that YadA binds to host cells, particularly phagocytes, via a bridging ECM molecule, which then allows the Ysc system to dock on the host cell surface and inject the Yops (Figure 5A) [229,233]. The host cell receptors involved in YadA-mediated Ysc docking are suggested to be β_1_ and α_V_ integrins [229,232]. Indirect binding of YadA to these receptors via an ECM molecule activates similar downstream signalling events as the direct binding of InvA [234]. In *Y. pestis*, which lacks both InvA and YadA, other proteins, such as the small β-barrel protein Ail, can aid in docking the T3SS [235,236]. Similarly, even in the absence of InvA and YadA, *Y. enterocolitica* and particularly *Y. pseudotuberculosis* are able to inject Yops into leukocytes, though at much reduced rates [230,232]. Ail and other adhesins may account for this low-level of Yop injection [226,237].

### 5.3. The Ysc-Yop T3SS

Like YadA, *ysc* is encoded by the pYV plasmid and upregulated at 37 °C [238]. The main functions of Ysc-Yops are to prevent phagocytosis of the bacteria and dampen immune responses. Though in vitro *Yersiniae* can inject Yops into epithelial cells and fibroblasts [229], the main targets for Yop injection in vivo appear to be leukocytes, particularly myeloid immune cells such as neutrophils and macrophages [230,239]. However, in the spleen, also B cells, and to a lesser degree T cells, can be targeted [230,239].

The effect of Yop injection is manifold. There are ten Yops (Table 1): three (LcrV, YopB and YopD) are involved in forming the tip of the injectisome and the translocon pore in the host cell membrane [240]. Another Yop, YopK (YopQ in *Y. enterocolitica*) is secreted in low amounts and appears to be involved in regulation of the secretion of other effector Yops [29]. The remaining Yops can be considered the true effectors with cytotoxic activities (Table 1). Four Yops (YopE, YopH, YopT and YopO, also called YpkA in *Y. pseudotuberculosis* and *Y. pestis*) are cytoskeletoxins that disrupt the actin cytoskeleton preventing the formation of phagocytic cups. Two of them, YopE and YopT, target Rho family GTPases [57]. YopE is a GTPase-activating factor that stimulates Rac1, RhoA, RhoG and also Cdc42 to hydrolyze their GTP cofactor, thus inactivating these proteins and preventing downstream actin polymerisation [94]. YopT prevents Rho family signalling by a different mechanism: YopT is a cysteine protease that cleaves the C-termini of Rho family members, thus separating them from their isoprenyl membrane anchor [98]. Following cleavage by YopT, the Rho proteins diffuse away from the membrane and their interaction partners, preventing signal propagation. The other two cytoskeletoxic Yops target phosphorylation of cytoskeletal signalling molecules. YopH is a potent and promiscuous phosphotyrosine phosphatase that targets several signalling molecules, especially those involved in the formation of focal adhesion, such as FAK and Cas, which are important for integrin signalling [95]. By contrast, YopO is a serine/threonine kinase that targets regulators of actin dynamics, thus preventing actin polymerisation [241].

The two other effectors have anti-inflammatory and pro-apoptotic effects; these are YopM and YopJ (YopP in *Y. enterocolitica*) [57]. YopJ is an acetyltransferase that modifies several kinases involved in signal transduction pathways activating the pro-inflammatory transcription factor NF-B and MAP kinases [207]. YopM is a leucine-rich repeat protein and the only Yop that appears to lack enzymatic activity. The molecular mechanism of how YopM exerts its anti-inflammatory function has remained elusive for a long time. In host cells, YopM can localize to both the cytosol and the nucleus. Interaction partners include the kinases RSK and PRK; however, the effect of these interactions is unclear. YopM also inhibits caspase-1, which might prevent pyroptosis of macrophages [72]. Recent evidence shows that YopM prevents the activation of Pyrin inflammasomes, which are induced by YopJ [242]. Although its mechanism still remains to be fully elucidated, YopM is a potent anti-inflammatory molecule that plays an important role in establishing infection.

In the *Yersiniae*, the T5SS (YadA or InvA) works together with the T3SS to inject Yops into phagocytes, allowing the bacteria to evade innate immune responses and persist extracellularly in lymphatic tissues. The association between YadA and the T3SS is yet another example of how the two secretion systems act together to promote pathogenesis by avoiding phagocytosis through disruption of the host immune cell cytoskeleton. 

## 6. Conclusions

The enterobacteria are a remarkably versatile group of pathogens. They display very different lifestyles, even between closely related bacteria. This is exemplified by the many pathovars of *E. coli*, including *Shigella* spp., which cause not only a range of gastrointestinal symptoms but also urinary tract infections, sepsis and meningitis [243]. The recently diverged *Y. pseudotuberculosis* and *Y. pestis* have completely different lifestyles and disease mechanisms [244].

The enterobacteria are also different regarding cell invasion. We provide examples of both intra- and extracellular pathogens, as well as facultatively intracellular bacteria. However, it should be noted that the distinction between extracellular, intracellular and facultative intracellular lifestyles are not clear-cut. EPEC and EHEC, though generally considered extracellular pathogens, can in some cases invade host cells [245,246,247]. Similarly, the *Yersiniae* can also invade and replicate inside host cells, even immune cells. Conversely, *Shigella*, typically thought of as an intracellular bacterium, can escape phagocytosis [248].

Central to the pathogenesis of the species discussed in this review, is the T3SS, which leads to alterations in the cytoskeletal dynamics of target cells, either to promote invasion or to prevent phagocytosis. This is accomplished by injecting cytoskeletoxic effectors. However, in this review, we have tried to emphasize that T5SSs also play an essential role in mediating the cytoskeletal changes brought about by T3SSs effectors in these bacteria. For example, Yop translocation into host cells is accomplished by docking the T3SS injectisome on the target cell by YadA [232]. The close co-evolution of YadA and the Ysc T3SS is demonstrated by the correlation in length of the injectisome needle and the YadA stalk [231]. The intimin–Tir interaction is not only an adhesin–receptor interaction, but represents an essential step in pedestal formation: the network of intimin–Tir complexes leads to clustering of Tir and the downstream effects leading to actin polymerisation [146,147]. This is also a direct interaction between an inverse autotransporter (intimin) on the bacterial surface and an effector protein on the host cell surface (Tir), demonstrating the tight interplay between the T3SS and T5SS. Both are encoded on the same pathogenicity island, the LEE, showing how these systems are also coupled on a genetic level [126]. Similarly, IcsA and YadA are encoded on virulence plasmids together with their respective T3SSs, showing that these virulence factors have co-evolved [174].

It should be noted that a similarly close connection between the T3SS and T5SS is not universal. For instance, within the Enterobacterales, a counterexample to the interplay between these two secretion systems seen in EPEC, *Shigella* and *Yersinia* would be *Salmonella*. *Salmonella* encodes two T3SSs needed for host cell invasion and intracellular life [249]. It also produces a several T5SSs [250]. However, the T5SSs and T3SSs of *Salmonella* do not have a similarly intimate relationship as in the examples we have given, and the autotransporters of *Salmonella* are not required for efficient effector translocation or T3SS-dependent cell invasion. 

Although the cytoskeletoxic effects of the examples above are due to indirect effects of T5SSs, some T5SSs can themselves have a direct effect on the cytoskeleton of host cells. This is exemplified most clearly by the autotransporter EspC, which is translocated into host cells by the T3SS to elicit cytoskeletal changes, among other effects [136]. The fact the EspC is not only translocated by the T3SS but also plays a role in regulating effector secretion by the T3SS again demonstrates co-evolution between the two secretion systems [137]. InvA alone is sufficient to induce changes in the actin cortex leading to uptake of the bacteria, or even InvA-coated beads [220]. Further, YadA can mediate cell invasion by a similar mechanism [233,234]. IcsA, in addition to promoting effector secretion, also directly interacts with actin-polymerising proteins to enable intracellular ABM [117,187]. Though of the examples given, only EspC is a genuine toxin—the other T5SSs covered in this review mediate adverse effects on host cells through changes in actin dynamics. Examples of T5SSs that interact with actin from outside the Enterobacterales include BimA, an actin-polymerising TAA from *Burkholderia* spp. [251,252], and Sca2, a classical autotransporter that enables ABM in *Rickettsia* spp [253]. Thus, not only the classical T3SS effectors but also some T5-secreted proteins could be considered cytoskeletoxins. 

The close co-evolution of the T3SS and T5SS in the enteropathogenic *Yersiniae*, *Shigella* spp., EPEC and EHEC is a testament to the versatility and genetic opportunism exhibited by pathogenic bacteria, and the interplay between the T3SS and T5SS is a fruitful and ongoing area of research.

## Figures and Tables

**Figure 1 ijms-21-04102-f001:**
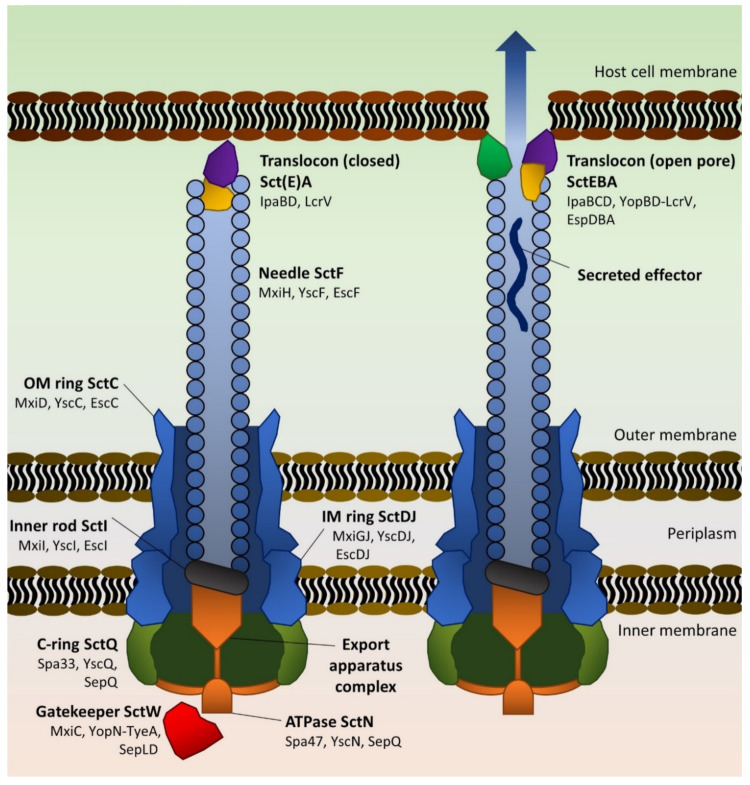
Schematic of type 3 secretion systems. The major structural rings (C-ring in olive, inner/outer membrane scaffold rings in blue) support the ATPase-containing export apparatus (orange), which is linked via an inner rod adaptor helix to the needle filament (grey oblongs and blue circles, respectively). Tip and gatekeeper proteins (purple, yellow, red) initially block the needle and prevent effector translocation (left) until the complex senses host cell contact—note that SctE (IpaB) has only been shown to play a role in blocking secretion in *Shigella*. Rearrangements then permit hydrophobic pore formation (purple, green) in the eukaryotic membrane and effector secretion (right). Proteins are not shown to scale. Adapted from [28,29,30,31].

**Figure 2 ijms-21-04102-f002:**
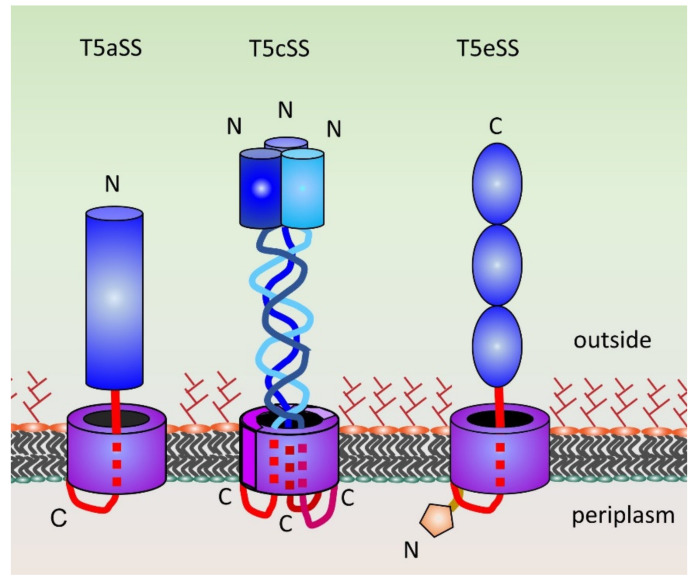
Schematic of type 5 secretion systems. Extracellular passengers are in blue, outer membrane β-barrels in purple and linkers connecting the β-barrel to the passenger in red. The passengers of T5aSSs are typically β-helical structures. T5cSSs or TAAs are formed by three identical chains. The passengers of TAAs like YadA consist of a coiled coil stalk and an N-terminal β-structured head domain [102], though more complicated architectures exist. T5eSSs, or inverse autotransporters, such as intimin and invasin have passengers consisting of tandem immunoglobulin-like domains often capped by a lectin-like domain. The transmembrane of all three classes consist of a 12-standed β-barrel (with three chains each contributing four β-strands in the case of TAAs) with the linker(s) contained within the hydrophilic lumen of the barrel [102]. The periplasmic LysM domain of inverse autotransporters is shown in orange. The positions of the N- and C-termini are indicated. The figure is adapted from [103].

**Figure 3 ijms-21-04102-f003:**
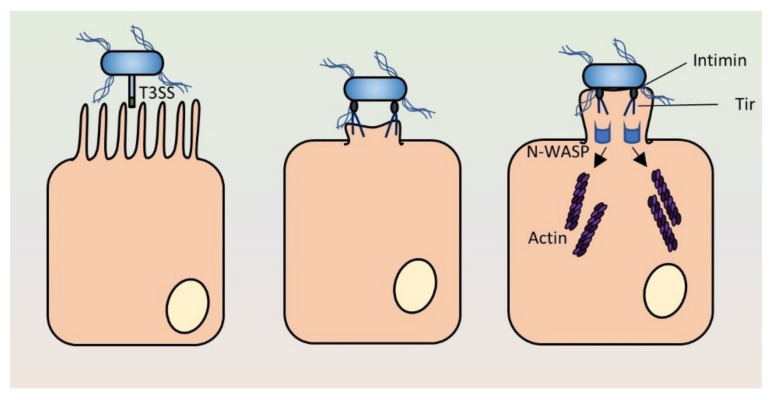
Schematic of the major steps in pedestal formation by attaching and effacing pathogens. The bacteria initially bind to the enterocytes via adhesins, allowing the Esc T3SS to inject effector proteins. These include Tir, which is inserted into the plasma membrane of the target cell where it binds to the inverse autotransporter intimin, present on the bacterial surface. The binding of intimin to Tir leads to intimate attachment and Tir clustering, resulting in recruitment of N-WASP, actin rearrangements and pedestal formation. Thus, the LEE-encoded T3SS and T5SS (intimin) act together to form the A/E lesion. Schematic of the major steps in pedestal formation by attaching and effacing pathogens. The bacteria initially bind to the enterocytes via adhesins, allowing the Esc T3SS to inject effector proteins. These include Tir, which is inserted into the plasma membrane of the target cell where it binds to the inverse autotransporter intimin, present on the bacterial surface. The binding of intimin to Tir leads to intimate attachment and Tir clustering, resulting in recruitment of N-WASP, actin rearrangements and pedestal formation. Thus, the LEE-encoded T3SS and T5SS (intimin) act together to form the A/E lesion.

**Figure 4 ijms-21-04102-f004:**
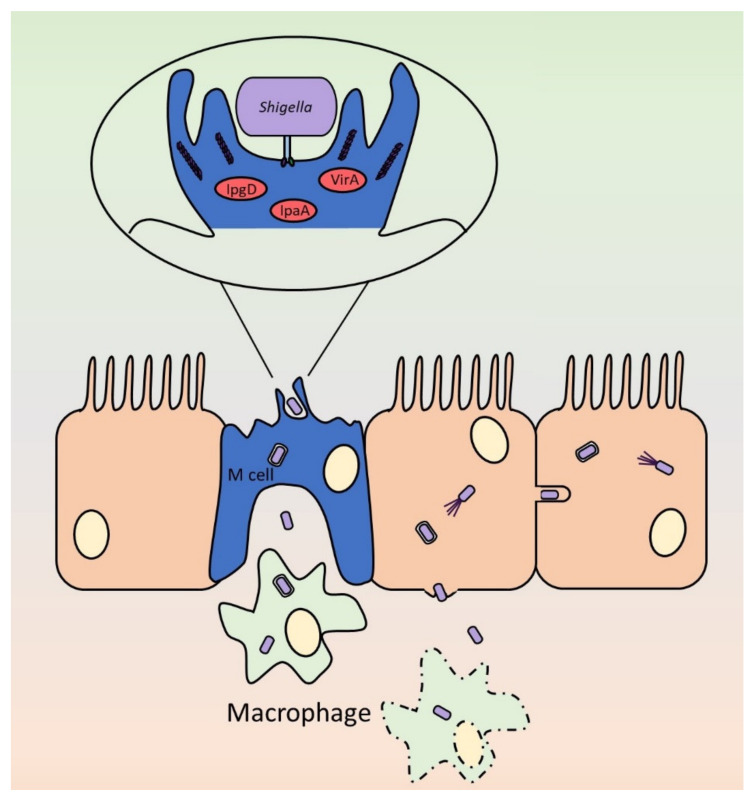
Schematic of the key steps in *Shigella* invasion and cell-to-cell spread through the intestinal epithelium. The bacteria firstly inject effectors via the T3SS into the host M cell to promote cytoskeletal remodelling. The actin-rich membrane ruffles engulf the bacteria followed by transcytosis and subsequent release from the M cell at the basolateral side. Here the bacteria are taken up by host macrophages where they induce rapid apoptosis through T3SS effectors, thus releasing the bacteria onto the basolateral surface of the epithelium. The bacteria then invade the epithelial cells via the T3SS. Following invasion, the bacteria recruit host actin to form “actin tails” via the autotransporter IcsA for intracellular dissemination for rapid lateral spread through the epithelium. The combination of the T3SS and T5SS allow the bacteria to invade and spread inter- and intracellularly.

**Figure 5 ijms-21-04102-f005:**
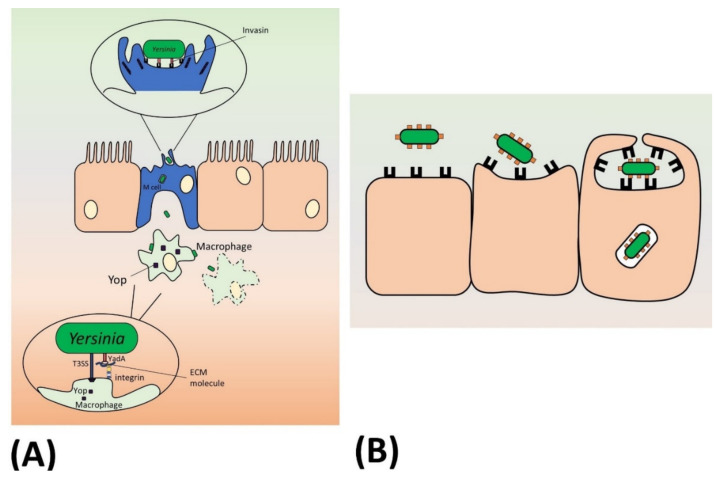
Schematic of enteropathogenic *Yersinia* pathogenesis. (**A**) In the distal ileum, *Yersiniae* bind to β_1_ integrin-expressing M cells via the inverse autotransproter invasin. This leads to internalisation of the bacteria and transcytosis through the M cell into the underlying Peyer’s patch, where the bacteria come into contact with immune cells such as neutrophils and macrophages. At this stage, the extracellular bacteria bind to immune cells via the TAA YadA. YadA binds to extracellular matrix (ECM) molecules such as collagen and fibronectin, which in turn are bound to host cell surface receptors such as integrins. The binding of YadA to the target cell through a bridging ECM molecule allows injection of Yops by the Ysc T3SS, which prevents phagocytosis and leads to apoptosis of the immune cells. In this way, the plasmid-encoded T3SS and T5SS (YadA) act together to evade phagocytosis and establish an extracellular infection. (**B**) Mechanism of invasin-mediated internalisation into host cells by the zipper mechanism. Invasin (orange) binds to β_1_ integrins on the host cell surface (black structures). This induces outside-in integrin signalling and changes to the actin cytoskeleton, which leads to engulfment of the bacteria by a zipper-like mechanism. YadA can also in some cases promote bacterial uptake by a similar mechanism via a bridging ECM molecule.

**Table 1 ijms-21-04102-t001:** Type 3 secretion system (T3SS) effector protein nomenclature.

Sct Name ^a^	*Shigella* and EIEC	*Yersinia* ^b^	EPEC and EHEC	Proposed Function	Reference
−	−	−	EspA	Filamentous bridge from needle to host	[22,59]
−	−	−	EspF	Phagocytosis inhibition, microvillus effacement, induction of apoptosis	[53,60]
−	−	−	EspF_U_	Actin pedestal formation (EHEC only)	[61]
			EspH	Phagocytosis inhibition via indirect Rho GTPase subversion	[53]
−	−	−	EspJ	Phagocytosis inhibition via ADP ribosylation of host kinases	[62]
−	−	−	EspZ	Regulation of effector translocation	[63]
−	IcsB	−	−	Autophagosome escape	[64]
−	IpaA	−	−	Binds/activates vinculin	[65]
SctE	IpaB	YopB	EspD	Translocon (pore forming)	[42,66,67]
SctB	IpaC	YopD	EspB	Translocon (pore forming; secreted effector)	[42,68,69]
SctA	IpaD	LcrV	−	Translocon (regulation of secretion)	[70,71]
−	IpaH	YopM	−	Immune subversion via E3 ubiquitin ligase activity	[72,73]
−	IpaJ	−	−	Golgi body fragmentation	[74]
−	IpgB1/2	−	EspM/EspT/Map	Host cytoskeleton remodelling via Rho GTPase activity	[75,76]
−	IpgD	−	−	Host cytoskeleton remodelling via ARF6 activation	[77]
−	−	−	LifA/Efa1 (lymphostatin)	Adhesin/toxin	[78]
−	OspB	−	−	Reduced cytokine production	[79]
−	OspC1	−	−	Increased neutrophil migration (bacterial access to submucosa)	[80]
−	OspC3	−	−	Pyroptosis inhibition via caspase-4 interference	[81]
−	OspD1	−	−	Regulation of secretion	[82]
−	OspD2	−	EspL ^c^	Pyroptosis inhibition via regulation of VirA translocation	[83,84]
−	OspD3 (ShET2)	−	EspL2 ^d^	Enterotoxin	[85,86]
−	OspE1/2	−	EspO1/2	Adhesin; actin stabilisation	[87,88]
−	OspF	−	−	Reduced cytokine production and increased neutrophil migration	[79,80]
−	OspG	YspK	NleH1/2	NF-κB attenuation via ubiquitin binding	[89]
−	OspI	CHYP	Cif	Inhibits TRAF6 activation via UBC13 deamidation	[90,91]
−	OspZ		NleE	Increased neutrophil migration (bacterial access to submucosa)	[92]
−	−	−	Tir	Actin pedestal formation	[93]
−	VirA	−	EspG	Autophagosome escape	[64]
−	−	YopE	−	Regulation; Rho GTPase activation	[94]
−	−	YopH	−	Immune subversion via phosphotyrosine phosphatase activity	[95]
−	−	YopP (YopJ)	−	Immune subversion via NF-κB attenuation	[96]
−	−	YopO (YpkA)	−	Prevents phagocytosis via phosphorylation of actin regulators	[97]
−	−	YopT	−	Immune subversion via Rho GTPase cleavage	[98]

^a^ Unified T3SS nomenclature as proposed by Wagner and Diepold [31]. ^b^ If different, Yersinia *enterocolitica* nomenclature is given first; *Y. pseudotuberculosis/pestis* nomenclature follows in parentheses. ^c^ OspD2 shows sequence homology to EspL but is not a cysteine protease. ^d^ EspL2 shows sequence homology to OspD3 but is not reported to have enterotoxin activity.

**Table 2 ijms-21-04102-t002:** Characterized autotransporters of pathogenic enterobacteria.

Organisms	Autotransporters	T5SS Subclass	Proposed Function	Reference
EHEC, EPEC				
	Cah	5a	Autoaggregation and biofilm formation	[107]
	EhaA	5a	Autoaggregation and biofilm formation	[108]
	EhaB	5a	Binding to extracellular matrix (ECM) molecules, biofilm formation	[109]
	EhaD ^a^	5a	Biofilm formation	[108]
	EhaG	5c	Binding to ECM molecules, biofilm formation	[110]
	EhaJ	5a	Binding to ECM molecules, biofilm formation	[111]
	EspC ^b^	5a	Regulation of T3SS secretion, cytotoxin	[112]
	EspP ^a^	5a	Regulation of T3SS, cleavage of coagulation and complement factors, formation of rope-like adhesive aggregates	[112]
	FdeC	5e	Binding to host cells and ECM molecules, autoaggregation and biofilm formation	[113]
	IatB	5e	Biofilm formation	[114]
	IatD	5e	Biofilm formation	[114]
	Intimin	5e	Binding to Tir; actin pedestal formation	[115]
	YeeJ	5e	Biofilm formation	[116]
EIEC, *Shigella* sp.				
	IcsA	5a	Adhesion to host cells; actin-based intracellular motility	[117]
	Pic	5a	Mucin degradation, serum resistance	[112]
	Sat ^c^	5a	Cytotoxin	[112]
	SepA ^c^	5a	Disruption of apical pole of polarized epithelial cells, facilitation of invasion	[112]
	SigA ^c^	5a	Cytotoxin	[112]
*Yersinia enterocolitica*, *Y. pseudotuberculosis*				
	Invasin (InvA)	5e	Binding to β_1_ integrins; invasion of M cells	[118]
	Ifp (InvB) ^d^	5e	Binding to and invasion of epithelial cells	[119]
	InvC	5e	Binding to epithelial cells	[120]
	InvD ^d^	5e	Binding to Fab fragment of immunoglobulins, binding to B cells	[121]
	YadA	5c	Binding to ECM molecules and serum components; host cell adhesion and immune evasion	[122]
	YapC ^d^	5a	Autoaggregation and biofilm formation, binding to host cells	[123]
	YapE	5a	Autoaggregation, binding to host cells	[124]
	YapV ^d^	5a	Binding to N-WASP and ECM molecules	[125]

^a^ Only in EHEC; ^b^ only in EPEC; ^c^ only in *Shigella* spp.; ^d^ only in *Y. pseudotuberculosis.*

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
