# Peer review of "Staying out or Going in? The Interplay between Type 3 and Type 5 Secretion Systems in Adhesion and Invasion of Enterobacterial Pathogens"

_ijms, 2020, doi:10.3390/ijms21114102_

Round 1
Reviewer 1 Report
In this manuscript, Whelan and colleagues review literature on the type 3 and type 5 secretion systems (T3SS and T5SS), with a special focus on how these systems act synergistically to promote successful infections by Gram-negative enteropathogenic bacteria. The authors focus on the relationship between the T3SS and the T5SS in gastrointestinal pathogens, specifically various Escherichia coli, Shigella and Yersinia subspecies. The review aims to demonstrate the coordinated use of the T3SS and T5SS in intracellular, extracellular and facultative-intracellular lifestyles.
The review is well-written, the figures and tables are helpful and the structure of the review (staying out – going in – bit of both) is a nice idea. The main point of criticism is that the review contains relatively few references that convincingly support the idea that the T3SS and T5SS indeed show a closer relation than any two virulence factors involved in infection. This connection, which is the unique point of this review, should therefore be explored more thoroughly.
Major points:
- Although basic information about the T3SS and T5SS themselves is required for this review, it would benefit from a tighter focus on the molecular and functional interplay between the type 3 and 5 secretion systems. As an example, some sections, such as the transcriptional regulation of the Shigella T3SS (lines 379-397) and the Yersinia effectors (lines 591-637), are extremely detailed, but do not support the claim of a T3SS-T5SS network.
- The claim that the T5SS should be considered a “cytoskeletoxin” rather than an adhesion factor is not necessarily substantiated. Yes, it is required for infection, but so is basically any virulence factor (or any essential bacterial protein). As mentioned before, the special connection between the T3SS and T5SS is not made clear enough to support such a claim.
Specific points:
- Line 54: which non-Gram-negative diderm bacteria do the authors refer to?
- Line 75 (and other parts of the manuscript, e.g. line 109): the T3SS does not only depend on ATP, but also (and arguably mainly) on the PMF. See Wilharm et al, Infection and immunity (2004) or Erhardt et al, PLOS Genetics (2014); reviewed amongst others in Renault et al, Current Topics in Microbiology and Immunology (2019).
- Line 90: What is the minimal genetic size of a complete T3SS (also compare to line 213).
- The authors should use the common Sct nomenclature, especially when they are referencing such a large number of proteins at once (i.e. lines 124-137). See Hueck et al, MMBR (1998); Portaliou et al, Trends Biochem Sci (2016); Wagner and Diepold, Current Topics in Microbiology and Immunology (2020).
- Several issues with figure 1:
- Use of Sct nomenclature in parallel to the species-specific names would be helpful.
- The inner rod formed by SctI has recently been shown to be a one-ring washer structure rather than a longer rod (Torres-Vargas et al, Mol Micro 2019)
- What structure does the export apparatus correspond to?
- What is the relation of tip and gatekeeper in the initial block of secretion?
- Do the SctA proteins (IpaD, LcrV, EspA) really leave the translocon when in contact do the host cell? Please include references in this case.
- Conversely, what is the evidence for a role of the SctE proteins (IpaB, YopB, EspD) in the closed translocon? To the best of my knowledge, this differs between species.
- Is the scale of the secreted effector realistic?
- Line 164: Reference for “most widespread”?
- Line 195: “shown in orange”
- Line 199: EHEC is repeated twice.
- Line 266-272: this is a very interesting point that should be extended. What is the benefit of the outside action of EspC for the bacteria? How could such a double role have evolved? Could the importance of the two roles be investigated separately?
- Line 281: Part of the sentence seems to be missing (“Intimin is an adhesion…”)
- Line 330, “from”
- Line 332: “than”
- Line 334-335: “… where the pedestal glides along the surface of the cell.” Should this read “where the cell glides along the surface of the pedestal.”?
- Line 353: “itself” duplicated
- Lines 364-367 and later (e.g. line 477): it does not become clear why the milder infections caused by EIEC should represent a more “basic” or “primitive” evolutionary stage compared to the more “optimized” Shigella, as is at least hinted throughout the text.
- Figure 4 and other figures shold more clearly highlight T3SS-T5SS roles and interaction.
- Line 555 and in general: abbreviations should be mentioned upon their first occurrence in the text and not only in line 674. Abbreviations that are only used once (such as TAA in this line) might be left out completely.
- Lines 563-564: the reference shows that shortening needles and/or increasing YadA length leads to inefficient delivery. In contrast, increasing needle length or shortening YadA does not seem to impede secretion. This should be corrected in the text.
- Lines 661-663: this statement should be substantiated.
Author Response
In this manuscript, Whelan and colleagues review literature on the type 3 and type 5 secretion systems (T3SS and T5SS), with a special focus on how these systems act synergistically to promote successful infections by Gram-negative enteropathogenic bacteria. The authors focus on the relationship between the T3SS and the T5SS in gastrointestinal pathogens, specifically various Escherichia coli, Shigella and Yersinia subspecies. The review aims to demonstrate the coordinated use of the T3SS and T5SS in intracellular, extracellular and facultative-intracellular lifestyles.
The review is well-written, the figures and tables are helpful and the structure of the review (staying out – going in – bit of both) is a nice idea. The main point of criticism is that the review contains relatively few references that convincingly support the idea that the T3SS and T5SS indeed show a closer relation than any two virulence factors involved in infection. This connection, which is the unique point of this review, should therefore be explored more thoroughly.
Response:
Thank you for your comments. We have increased our emphasis on the close connection between the T3SS and T5SS in our example pathogens. We have also re-written our conclusion section to more strongly re-iterate our case. Please also see responses below.
Major points:
Although basic information about the T3SS and T5SS themselves is required for this review, it would benefit from a tighter focus on the molecular and functional interplay between the type 3 and 5 secretion systems. As an example, some sections, such as the transcriptional regulation of the Shigella T3SS (lines 379-397) and the Yersinia effectors (lines 591-637), are extremely detailed, but do not support the claim of a T3SS-T5SS network.
We have reduced these sections as the reviewer suggests. While the main aim of this review is to show how the T3SS and T5SS cooperate in EPEC, Shigella and Yersinia, we also wish to give an overview of pathogenesis, so we have not removed these sections entirely.
The claim that the T5SS should be considered a “cytoskeletoxin” rather than an adhesion factor is not necessarily substantiated. Yes, it is required for infection, but so is basically any virulence factor (or any essential bacterial protein). As mentioned before, the special connection between the T3SS and T5SS is not made clear enough to support such a claim.
Again, thank you for your input. We have now tried to state more clearly that not all T5SS can be considered cytoskeletoxins, but the examples we provide demonstrate that T5SSs can be directly involved in the alteration of host cytoskeletal processes for the benefit of the bacteria. We have re-written the conclusions to make these effects clearer. We have also added some non-enterobacterial examples in the conclusions to bolster the claim that certain other T5SSs have similar effects. However, we concede the fact that not all T5SSs have such an effect on cytoskeletal processes and note this fact in the conclusions. Similarly, we now point out that in some other model organisms, such as Salmonella, such a special relationship between T3SS and T5SS does not exist.
Specific points:
Line 54: which non-Gram-negative diderm bacteria do the authors refer to?
We mean mycobacteria; we have made this explicit on line 54.
Line 75 (and other parts of the manuscript, e.g. line 109): the T3SS does not only depend on ATP, but also (and arguably mainly) on the PMF. See Wilharm et al, Infection and immunity (2004) or Erhardt et al, PLOS Genetics (2014); reviewed amongst others in Renault et al, Current Topics in Microbiology and Immunology (2019).
Thank you for pointing this out. We have noted this on lines 75-76 and added a reference to the review.
Line 90: What is the minimal genetic size of a complete T3SS (also compare to line 213).
We have added this information (lines 97-98).
The authors should use the common Sct nomenclature, especially when they are referencing such a large number of proteins at once (i.e. lines 124-137). See Hueck et al, MMBR (1998); Portaliou et al, Trends Biochem Sci (2016); Wagner and Diepold, Current Topics in Microbiology and Immunology (2020).
Thanks you for pointing this out. We have added Sct nomenclature to Figure 1 and Table 1, and referred to T3SS proteins using this nomenclature in the text when not discussing a particular T3SS.
Several issues with figure 1:
Use of Sct nomenclature in parallel to the species-specific names would be helpful.
This has been added.
The inner rod formed by SctI has recently been shown to be a one-ring washer structure rather than a longer rod (Torres-Vargas et al, Mol Micro 2019) What structure does the export apparatus correspond to?
This has been changed as suggested.
What is the relation of tip and gatekeeper in the initial block of secretion?
This has been changed and a reference added.
Do the SctA proteins (IpaD, LcrV, EspA) really leave the translocon when in contact do the host cell? Please include references in this case.
They do not; the schematic has been changed accordingly.
Conversely, what is the evidence for a role of the SctE proteins (IpaB, YopB, EspD) in the closed translocon? To the best of my knowledge, this differs between species.
This has been changed in the schematic and a reference has been added to show this is the behaviour of IpaB.
Is the scale of the secreted effector realistic?
No, this has been made clear in the legend.
Line 164: Reference for “most widespread”?
A reference has been added to line 72.
Line 195: “shown in orange”
Corrected.
Line 199: EHEC is repeated twice.
Thanks you for noticing, the second one was corrected to EPEC.
Line 266-272: this is a very interesting point that should be extended. What is the benefit of the outside action of EspC for the bacteria? How could such a double role have evolved? Could the importance of the two roles be investigated separately?
Thank you for these suggestions. We have expanded this section on the extracellular regulatory role of EspC in the text. We also point out some other extracellular functions EspC may have. To our knowledge, there are no published studies or hypotheses for how the dual (extra- and intracellular) functions might have evolved, but this is in keeping with the observations that many autotransporters are multifunctional proteins.
Line 281: Part of the sentence seems to be missing (“Intimin is an adhesion…”)
Corrected, thank you for noticing.
Line 330, “from”
Corrected.
Line 332: “than”
Thank you for noticing, corrected.
Line 334-335: “… where the pedestal glides along the surface of the cell.” Should this read “where the cell glides along the surface of the pedestal.”?
In fact the entire pedestal moves along with the bacteria. We have clarified the sentence to make this more explicit.
Line 353: “itself” duplicated
Corrected, thank you for noticing.
Lines 364-367 and later (e.g. line 477): it does not become clear why the milder infections caused by EIEC should represent a more “basic” or “primitive” evolutionary stage compared to the more “optimized” Shigella, as is at least hinted throughout the text.
We have added further explanations for why EIEC is less virulent than Shigella to lines 392-393. The regulatory differences that probably contribute to different disease outcomes are explained on lines 498-500.
Figure 4 and other figures shold more clearly highlight T3SS-T5SS roles and interaction.
We have added further explanation to the figure legends highlighting the roles of T3SS and T5SS in the different stages of pathogenesis.
Line 555 and in general: abbreviations should be mentioned upon their first occurrence in the text and not only in line 674. Abbreviations that are only used once (such as TAA in this line) might be left out completely.
TAA was introduced on line 180 and used several times in that section. We have avoided introducing unnecessary abbreviations; all the abbreviations in the list are used at least twice.
Lines 563-564: the reference shows that shortening needles and/or increasing YadA length leads to inefficient delivery. In contrast, increasing needle length or shortening YadA does not seem to impede secretion. This should be corrected in the text.
Thank you for noticing, we have corrected this.
Lines 661-663: this statement should be substantiated.
This whole section has been revised and references have been added.
Reviewer 2 Report
The review comprehensively summarizes the state of the art on the interplay between type III and type V secretion systems in E. coli, Shigella and Yersinia during their pathogenesis. Differences in behavior are highlighted using specific examples from the respective bacteria. The secretion systems, their action, and interaction are described thoroughly and with great attention to detail. The authors mention coevolution of T3SS and T5SS in the organisms they discuss. However, this point is not elaborated further in the present review or the references given therein. Since this is, to my knowledge, one of the first reviews systematically discussing the symbiosis of these two secretion systems, the authors may wish to give more details on this, if available. Furthermore, while many well-studied organisms have both T3SS and T5SS, only a select few demonstrate the interplay discussed in this review. The authors may wish to further discuss what differentiates the organisms and systems presented here from other bacteria possessing both SSs. Minor remarks Line 27, 54, ... : Gram should be capitalized and never hyphenated when used as Gram stain; gram negative and gram positive should be lowercase and only hyphenated when used as a unit modifier. Line 98 (Figure 1), as well as chapter 2.1.2: unless discussing proteins from specific T3SSs, please use the unified nomenclature, as described in Wagner S., Diepold A. (2020) A Unified Nomenclature for Injectisome-Type Type III Secretion Systems. In: . Current Topics in Microbiology and Immunology. Springer, Berlin, Heidelberg (https://doi.org/10.1007/82_2020_210) Line 171: T5eSSs inverse or autotransporters Chapter 3.2 and later: Intimin should not be capitalized per se. Line 330/331: Inconsistent capitalization of EspFu vs EspFU Line 364: cell-to-cell --> cell to cell. Line 390: Superfluous comma after FNR Line 442: et al. Line 577: "Like YadA, ysc encoded by the pYV plasmid and upregulated at 37 °C [233]." - this sentence no verb.
Author Response
Reviewer #2:
The review comprehensively summarizes the state of the art on the interplay between type III and type V secretion systems in E. coli, Shigella and Yersinia during their pathogenesis. Differences in behavior are highlighted using specific examples from the respective bacteria. The secretion systems, their action, and interaction are described thoroughly and with great attention to detail. The authors mention coevolution of T3SS and T5SS in the organisms they discuss. However, this point is not elaborated further in the present review or the references given therein. Since this is, to my knowledge, one of the first reviews systematically discussing the symbiosis of these two secretion systems, the authors may wish to give more details on this, if available.
Thank you for your suggestion, which is along the same lines as that made by reviewer #1. We have expanded our conclusions section and elaborated on the way the two secretion systems cooperate. We have also included a more focused summary on the co-evolution of the systems.
Furthermore, while many well-studied organisms have both T3SS and T5SS, only a select few demonstrate the interplay discussed in this review. The authors may wish to further discuss what differentiates the organisms and systems presented here from other bacteria possessing both SSs.
Thank you for pointing this out. We have now done so in our concluding section, where we make the point that this interplay is not universal, but largely restricted to the examples we give (though of course more may be discovered in the future).
Minor remarks Line 27, 54, ... : Gram should be capitalized and never hyphenated when used as Gram stain; gram negative and gram positive should be lowercase and only hyphenated when used as a unit modifier.
Here we disagree. Gram is a name and should therefore be capitalised in all instances.
Line 98 (Figure 1), as well as chapter 2.1.2: unless discussing proteins from specific T3SSs, please use the unified nomenclature, as described in Wagner S., Diepold A. (2020) A Unified Nomenclature for Injectisome-Type Type III Secretion Systems. In: . Current Topics in Microbiology and Immunology. Springer, Berlin, Heidelberg (https://doi.org/10.1007/82_2020_210)
We have done as suggested, as also urged by reviewer #1. The nomenclature has been changed in both the figure and the text.
Line 171: T5eSSs inverse or autotransporters
Corrected.
Chapter 3.2 and later: Intimin should not be capitalized per se.
Thank you for noticing; we have corrected this.
Line 330/331: Inconsistent capitalization of EspFu vs EspFU
Again, thank for noticing, we have change this to EspFU in all cases.
Line 364: cell-to-cell --> cell to cell.
Changed.
Line 390: Superfluous comma after FNR
Corrected.
Line 442: et al.
Corrected.
Line 577: "Like YadA, ysc encoded by the pYV plasmid and upregulated at 37 °C [233]." - this sentence no verb.
Thank you for noticing, we have added one.
Reviewer 3 Report
In the present manuscript the authors describe the pathogenesis of Enteribacteriaceae and Yersiniaceae and more specifically the type 3 and type 5 secretion systems of the bacteria and how these are utilized in the pathogenesis. The differences of in virulence factors and proteins playing the part in virulence are compared. The manuscript is well written and clear although the systems described here contain a plenty of proteins and mechanisms.
There are some concerns need to be addressed.
Introduction:
Line 39:
Shigella species can not be separated from E. coli based on the gene coding 16S rRNA. Describe which kind of diagnostic tools are used to detect Shigella species. Now you just describe that there are diagnostic challenges.
2.1.1. Overview
Line 87:
Y. enterocolitica and Y. pseudotuberculosis are mentioned here for the first time. If the name of bacterial species is mentioned for the first time, the whole name (genus name) should be opened. Here you are using only Y. to describe Yersinia. Please make sure that this will be corrected in the case of all bacterial names included in the manuscript.
Figure1:
The color difference between blue and purple circles is not detectable. Please use more clear color difference to separate these.
What is the function of violet/purple molecule in the middle of green C-ring? There is nothing said about this molecule in figure legend. If it has different function compared to Translocon protein, the color should be different.
Figure 2:
The color difference between blue (extracellular passenger) and purple (β-barrels) is not detectable. Please use more clear differences between the colors. Linkers seem to be red rather than brown in the figure. Please correct the colors so that they match between figure and figure legend.
Line 188: “T3cSS” ??? I should be T5cSS? Right? Please correct that one.
3.1. The Esc-Esp T3SS
You describe a plenty of molecules and proteins here. This chapter was unclear because of the amount of protein names. I suggest that you add one more table to help reader to follow the text.
3.3. Pedestal formation
Line 332:
“then” should be “than” at the end of the line. Am I right?
Figure 5:
You have used ECM to describe extracellular matrix proteins in the figure legend. ECM abbreviation is explained in line 558 which appears later in the manuscript. If you are not familiar with the abbreviation this is confusing. Please describe it also in the figure legend.
Please add some description of figure 5 B. There is almost nothing said about the figure.
Line 534: Please remove the extra dot.
Author Response
In the present manuscript the authors describe the pathogenesis of Enteribacteriaceae and Yersiniaceae and more specifically the type 3 and type 5 secretion systems of the bacteria and how these are utilized in the pathogenesis. The differences of in virulence factors and proteins playing the part in virulence are compared. The manuscript is well written and clear although the systems described here contain a plenty of proteins and mechanisms.
There are some concerns need to be addressed.
Introduction:
Line 39:
Shigella species can not be separated from E. coli based on the gene coding 16S rRNA. Describe which kind of diagnostic tools are used to detect Shigella species. Now you just describe that there are diagnostic challenges.
We address this issue later in section 4, where we point out that the metabolic differences between Shigella and EIEC allow differential diagnosis (see lines 394-396).
2.1.1. Overview
Line 87:
- enterocolitica and Y. pseudotuberculosis are mentioned here for the first time. If the name of bacterial species is mentioned for the first time, the whole name (genus name) should be opened. Here you are using only Y. to describe Yersinia. Please make sure that this will be corrected in the case of all bacterial names included in the manuscript.
Thank you for noticing, we have corrected Yersinia and checked the other names for consistency.
Figure1:
The color difference between blue and purple circles is not detectable. Please use more clear color difference to separate these.
We have changed the colours in the figure and amended the legend.
What is the function of violet/purple molecule in the middle of green C-ring? There is nothing said about this molecule in figure legend. If it has different function compared to Translocon protein, the color should be different.
The colors have been changed.
Figure 2:
The color difference between blue (extracellular passenger) and purple (β-barrels) is not detectable. Please use more clear differences between the colors. Linkers seem to be red rather than brown in the figure. Please correct the colors so that they match between figure and figure legend.
The color differences have been made clearer.
Line 188: “T3cSS” ??? I should be T5cSS? Right? Please correct that one.
Indeed, corrected.
3.1. The Esc-Esp T3SS
You describe a plenty of molecules and proteins here. This chapter was unclear because of the amount of protein names. I suggest that you add one more table to help reader to follow the text.
We have slightly reduced this section and refer the reader to table 1, where the effectors are listed.
3.3. Pedestal formation
Line 332:
“then” should be “than” at the end of the line. Am I right?
Indeed you are. We have corrected this.
Figure 5:
You have used ECM to describe extracellular matrix proteins in the figure legend. ECM abbreviation is explained in line 558 which appears later in the manuscript. If you are not familiar with the abbreviation this is confusing. Please describe it also in the figure legend.
We have added the explanation for ECM to Table 1, Figure 5 in addition to the main text.
Please add some description of figure 5 B. There is almost nothing said about the figure.
We have added additional text.
Line 534: Please remove the extra dot.
We are not sure what the reviewer is referring to here; we could not see an additional dot on this line or other lines nearby.
Round 2
Reviewer 1 Report
The authors have addressed most of the comments raised earlier.
I only have two minor issues with the updated Figure 1:
- The location of the export apparatus and SctI are a bit misleading. A correction, based on latest structural information (e.g. Fig. 1 in Torres-Vargas et al., https://doi.org/10.1111/mmi.14327) would ensure that this central figure of the review is not based on outdated information).
- SctA is still not included in the right structure (text in the figure: "SctBE" and species-specific names), this should be changed.
Author Response
Thank you for drawing our attention to this; we have modified the figure and uploaded a revised version.